# [Re] A Cluster-based Approach for Improving Isotropy in Contextual Embedding Space

## Reproducibility Summary

**Scope of Reproducibility**

The authors of the paper, which we reproduced, introduce a method that is claimed to improve the isotropy (a measure of uniformity) of the space of Contextual Word Representations (CWRs), outputted by models such as BERT or GPT-2. As a result, the method would mitigate the problem of very high correlation between arbitrary embeddings of such models. Additionally, the method is claimed to remove some syntactic information embedded in CWRs, resulting in better performance on semantic NLP tasks. To verify these claims, we reproduce all experiments described in the paper.

**Methodology**

We used the authors' Python implementation of the proposed cluster-based method, which we verified against our own implementation based on the description in the paper. We re-implemented the global method based on the paper from Mu and Viswanath [11], which the cluster-based method was primarily compared with. Additionally, we re-implemented all of the experiments based on descriptions in the paper and our communication with the authors.

**Results**

We found that the cluster-based method does indeed consistently noticeably increase the isotropy of a set of CWRs over the global method. However, when it comes to semantic tasks, we found that the cluster-based method performs better than the global method in some and worse in other tasks, or that the improvements are within margin of error. Additionally, the results of one side experiment, which analyzes the structural information of CWRs, also contradict the authors' findings for the GPT-2 model.

**What was easy**

The described methods were easy to understand and implement, as they rely on PCA and K-Means clustering.

**What was difficult**

There were many ambiguities in the paper: which splits of data were used, the procedures of the experiments were not described in detail, some hyperparameters values were not disclosed. Additionally, running the approach on big datasets was too computationally expensive. There was an unhandled edge case in the authors' code, causing the method to fail in rare cases. Some results had to be submitted online, where there is a monthly limit of submissions, causing delays.

**Communication with original authors**

We exchanged many e-mails with the authors, which were very responsive and helpful in describing the missing information required for reproduction. In the end, we still could not completely identify the sources of some remaining discrepancies in the results, even after ensuring the data, preprocessing and some other implementation details were the same.

Submitted to ML Reproducibility Challenge 2021. Do not distribute.

# 1    Introduction

Embeddings from popular contextual NLP models such as BERT [6], GPT-2 [12], RoBERTa [8], etc. suffer from the so-called representation degeneration problem [7], where the individual tokens' embeddings form an anisotropic cone-like shape in the embedding space. This means that even unrelated words can have excessively positive correlations.

Methods which study and attempt to improve the isotropy (a measure of uniformity) of the space on a global level (e.g. [11]) have been predominantly used so far to tackle this problem. However, due to the clustered structure of these Contextual Word Representations (CWRs), the authors of the chosen paper [13] propose a local, cluster-based method, which could further improve on the existing global approaches.

Apart from further improving isotropy, the method supposedly also removes some local structural and syntactic information within the clusters, improving the CWRs performance on semantic tasks.

# 2    Scope of reproducibility

Throughout the paper, the authors use contextual embeddings of three models to support their claims: BERT, RoBERTa and GPT-2. Various datasets are used to generate these contextual embeddings, which are then enhanced with the proposed method, evaluated and used to support claims about the performance of the method. Specifically, these claims are:

- Claim 1: The cluster-based method outperforms the baseline and global method, in all cases in terms of isotropy of CWRs as well as in almost all cases in terms of Spearman correlation performance, on 7 Semantic Textual Similarity (STS) datasets.

- Claim 2: A wide and shallow Multi-Layer Perceptron (MLP) performs the best in terms of accuracy on all 6 chosen binary classification tasks from the GLUE [16] and SuperGLUE [15] benchmarks, when trained on BERT emebeddings which were enhanced by the cluster-based approach.

- Claim 3: A MLP described as in Claim 2 also converges to an optimum in fewer epochs, when the embeddings are enhanced by the cluster-based approach.

- Claim 4: Removing dominant directions from CWRs of punctuations and stop words in sentences with the same syntactic structure (same group) results in fewer nearest neighbors of the CWRs being from the same group, as syntactic information is discarded.

- Claim 5: The cluster-based approach brings together verbs which have the same meaning (sense) but different tense as seen in the SemCor corpus, by decreasing the average euclidean distance between their CWRs, relative to the distance between verbs in the same tense but with a different sense.

In our reproduction, we verify all the listed claims by reproducing all the related experiments. **Claims 1 and 2 are the most important ones** as they directly address the performance of the cluster-based method, while Claims 3, 4 and 5 are essentially attempted explanations of different side effects of the proposed method.

In addition to these claims, the authors analyze the effect of the number of clusters in the K-Means algorithm on isotropy as well as evaluate the layer-wise isotropy of the contextual models. We have also reproduced these, purely statistical experiments for the sake of completeness of our reproduction.

# 3    Methodology

The paper referenced a Github repository [1], in which we found a single Jupyter notebook with the implementation of the cluster-based method, the isotropy metric, as well as an example of evaluating the isotropy and Spearman correlation performance on the STS-B dataset. We first re-implemented the cluster-based method and verified that it works the same way – however in the end we used the authors implementation due to its slightly better runtime. There was an unhandled edge case in the original implementation however – if fewer embeddings belonged to some cluster than the number of PCs to be removed, the original implementation would result in an out-of-bounds exception. We fixed this

---

[1]https://github.com/Sara-Rajaee/clusterbased_isotropy_enhancement/

by repeating the clustering step until each cluster was sufficiently represented. The method uses the Scipy library for K-Means clustering and ScikitLearn for PCA.

As the global method is simply a special case of the cluster-based method with the number of clusters $k = 1$, its re-implementation was trivial.

We did however have to re-implement all of the experiments only from their descriptions in the paper and based on the help we got from our correspondence with the authors. We did not require a GPU for any of our experiments.

### 3.1 Model descriptions & hyperparameters

For the contextual models, we used the Transformers library and the default pre-trained weights were used (specifically the casings *bert-base-uncased*, *gpt2* and *roberta-base*). These models all output 768-dimensional embeddings at each of their 12 layers.

As reported in the original paper, the hyperparameters of the global and local, cluster-based approach were set for each model separately, as seen in Table 1. These values were used for all experiments.

| Model | k | Removed PCs (local) | Removed PCs (global) |
|---|---|---|---|
| BERT | 27 | 12 | 15 |
| GPT-2 | 10 | 30 | 30 |
| RoBERTa | 27 | 12 | 25 |

Table 1: The number of clusters for the K-Means clustering local method (k) and number of top principal components removed for both the local and global method, for each contextual model.

When it comes to GLUE and SuperGLUE binary classification tasks, the contextual embeddings were used to train a fully-connected MLP. It's structure remains the same across all tasks, using the hyperparameters communicated to us by the authors. Specifically, for a single data sample (which is either a sentence or a pair of sentences), we only consider the first 64 tokens' representations, which we flatten into a vector of length $64 \times 768$, which represents our input layer. The next layer is a 100-dimensional hidden layer with ReLU activation, followed by the output layer – a single neuron with sigmoid activation. The MLP is trained using binary cross-entropy loss and uses the Adam optimizer with step size 0.005, for a maximum of 10 epochs. The reported results are based on the model which achieves the best validation set score.

For the experiment where we analyze the CWRs of punctuations and stop words, we use the K-nearest-neighbor implementation by ScikitLearn with $k = 6$, which is exactly the number of possible neighbors from the same structural group (we only use the first CWR of the respective punctuation or stop word in a sentence). We then calculate the relative part of nearest neighbors belonging to the same group for each individual embedding and average the results. Note that each stop word or punctuation type (e.g. comma) is analyzed separately and the search is performed only amongst CWRs of the same type.

Lastly, for the verb tense experiment, we consider verbs with multiple meanings (senses) and in two tenses – present simple and past simple (e.g. "say" and "said" correspond to the same verb in different tenses by our definition). Then, for each verb, we calculate all possible euclidean distances between representations of same tense and same meaning, same tense and different meaning, different tense and same meaning. We then finally average across all distances at the lowest level of hierarchy. We repeat the calculation for the representations enhanced by the cluster-based method.

### 3.2 Datasets

For the main experiment on which Claim 1 in Section 2 is based, 7 Semantic Textual Similarity (STS) datasets were used. The STS-2012 to STS-2016 [4, 5, 2, 1, 3] as well as STS-B are available at: `https://ixa2.si.ehu.eus/stswiki/index.php/Main_Page` , while the SICK-R [9] dataset is available at: `https://marcobaroni.org/composes/sick.html`. Individual data samples of these datasets are comprised of two sentences, and their semantic similarity/relatedness score, which is a real value on the scale from 0 to 5. In Table 2, the total number of data samples for each dataset after filtering is seen. Note that only the English test splits were used, as in the original paper. Four of the seven datasets had some badly encoded samples (no more than 10), which we simply discarded, after preliminary

testing which showed that they do not noticeably affect the results. The two sentences of each sample were sent through the contextual models separately.

| Dataset | Test data samples |
|---------|-------------------|
| STS-2012 | 3101 |
| STS-2013 | 2250 |
| STS-2014 | 3746 |
| STS-2015 | 2983 |
| STS-2016 | 1162 |
| STS-B | 1095 |
| SICK-R | 9840 |

Table 2: The number of used data samples in each STS dataset.

| Task | Train split (used / total) | Validation split (used / total) | Test split | Total (used) |
|------|----------------------------|----------------------------------|------------|--------------|
| RTE | 2490 / 2490 | 277 / 277 | 3000 | 5767 |
| CoLA | 8551 / 8551 | 1043 / 1043 | 1063 | 10657 |
| SST-2 | 7000 / 67349 | 872 / 872 | 1821 | 9693 |
| MRPC | 3668 / 3668 | 408 / 408 | 1725 | 5801 |
| WiC | 5428 / 5428 | 638 / 638 | 1400 | 7466 |
| BoolQ | 6000 / 9427 | 1500 / 3270 | 3245 | 10745 |

Table 3: The number of used data samples in each GLUE/SuperGLUE task.

For the classification experiment on which Claim 2 in Section 2 is based, a selection of tasks (datasets) from GLUE [16] (`https://gluebenchmark.com/`) and SuperGLUE [15] (`https://super.gluebenchmark.com/`) were used. In some cases, data samples were composed of pairs of sentences, while in others, a single sentence was given. In the first case, the pairs of sentences were encoded together, by concatenating their tokens and adding special tokens in the following way: [CLS]<sentence1>[SEP]<sentence2>[SEP]. The embeddings of these special tokens were also considered by the MLP classifier. Note that for the purpose of this experiment, we first merged the train, validation and test splits before applying the global or local enhancement method, as did the authors originally. Due to the big size of SST-2 and BoolQ datasets, we had to limit the size of training and/or validation splits by random sub-sampling. The number of samples for each task are seen in Table 3. We found that $10745 \times 64$ was near the maximum number of embeddings that we could affoard to run PCA on, given our hardware.

For the punctuation / stop word experiment, the authors provided a dataset based on Ravfogel et al. [14] (available at `https://nlp.biu.ac.il/~ravfogs/resources/syntax_distillation/`) which consists of 150000 groups of 6 sentences, where sentences from each group have the same syntactic structure but different semantics. For each of the tokens of interest separately ("the", "of", "," and "."), we randomly sampled 200 groups, where each group contained at least one appearance of the token per sentence.

For the verb tense experiment, we used the SemCor corpus [10], available at `http://web.eecs.umich.edu/~mihalcea/downloads.html#semcor`. Out of over 30000 sentences, we used 11838 of them, which contained the verbs we were interested in. Specifically, these were verbs that appeared in present and past tense and also occurred in at least 2 different senses at least 10 times.

The analysis of layer-wise isotropy and the number of clusters in K-Means is done on the STS-B dev split.

### 3.3 Experimental setup and code

The code of our reproduction is available at `https://anonymous.4open.science/r/isotropy_reproduction-CE64/`.

The isotropy measure (as defined in the original paper), Spearman performance (which is just the Spearman coefficient multiplied by 100) and accuracy were the main metrics used to evaluate our experiments. In order to evaluate the uncertainty in some of the main results, we resorted to bootstrap as well estimation of variance across multiple re-runs of procedures containing stochasticity (e.g. initial positions of centroids in K-Means, initial weights of MLP classifiers).

### 3.4 Computational requirements

The experiments were reproduced on a sytem with the 8-core, 16-thread Ryzen 3700x processor, 16GB of RAM and RTX3060Ti GPU (which was not explicitly used for any experiment).

On a set of 30000 768-dimensional embeddings, the global method ran for 12.5 seconds and the local, cluster-based method for 14 seconds. On a bigger set of 200000 embeddings, the global method ran for 98.9 seconds and the local method ran for 79.8 seconds. In addition, the local method requires a lot less memory at once, as it performs PCA for each cluster of embeddings separately.

148 The training of MLP classifiers for the classification experiments required no more than a minute on average.

## 4  Results

150 The reproduced results support some of the claims of the original paper. Specifically, the cluster-based method indeed
151 consistently outperforms the global and baseline in terms of isotropy. However, when it comes to Spearman performance
152 on Semantic Textual Similarity tasks, the local method performs better than the global method on some datasets and
153 worse on others. Similar is true for the classification tasks, where the difference in performance is mostly within
154 margin of error. Analyzing verb tense, the Claim 5 from Section 2 is fully supported by our reproduction, while some
155 discrepancies are observed when it comes to Claim 4.

### 4.1  Results reproducing original paper

#### 4.1.1  Semantic Textual Similarity experiment

158 In this section we address Claim 1 from Section 2. In Figure 1 we plot the Spearman correlation performance for each
159 method, contextual model and STS dataset. Due to the random nature of K-Means, we repeat the experiment with
160 the local method 5 times. We plot the results for each of the five repetitions individually. Additionally, we report the
161 averages of these five repetitions in Table 4. Compared to the numbers in Table 2 of the original paper, our results are
162 slightly more pessimistic. Embeddings enhanced by the local method perform noticeably better than those, enhanced
163 by the global method, on some datasets and worse on others. There are also many cases where the difference in
164 performance is within margin of error.

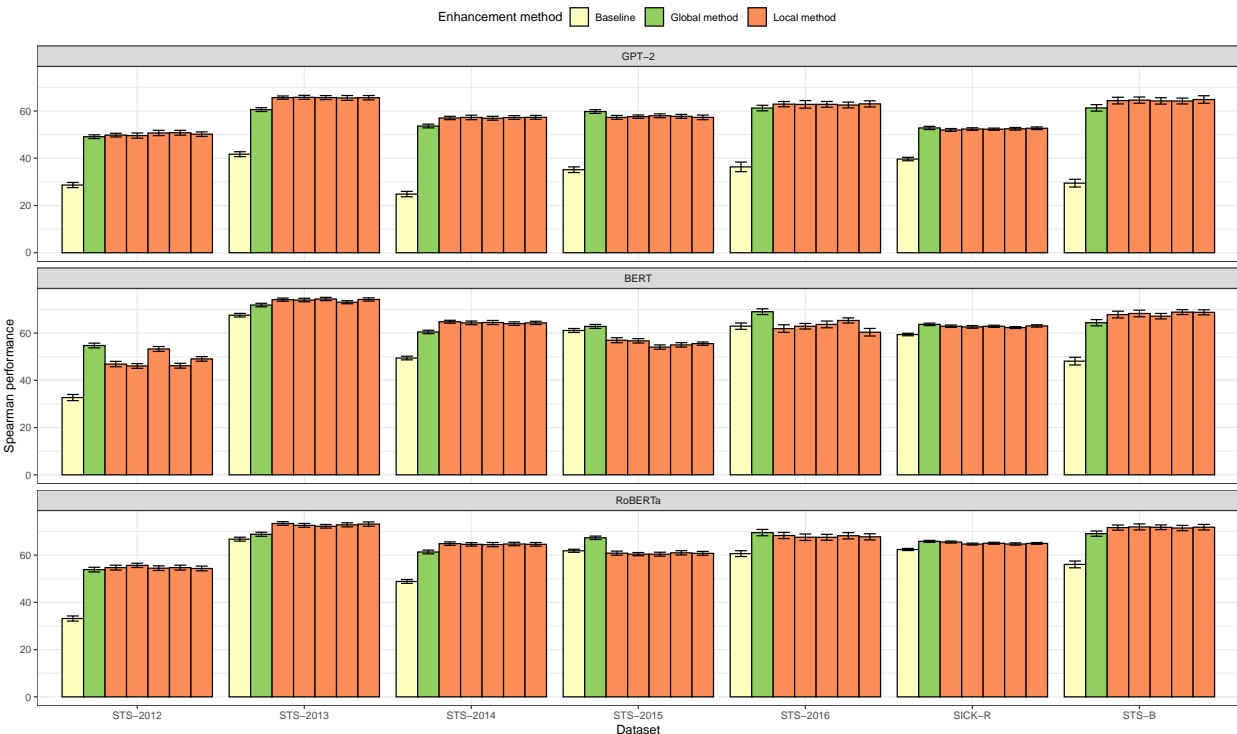

Figure 1: Spearman correlation performance on STS tasks. The error bars mark $\pm 1$ SE, based on 50 bootstrap
replications.

165 In Table 5 we report the isotropy values of CWRs from each of the STS datasets, for each contextual model and
166 enhancement method. These results support the original results achieved by the authors, as seen in Table 6 of the
167 original paper.

| Model | STS-2012 | STS-2013 | STS-2014 | STS-2015 | STS-2016 | SICK-R | STS-B |
|---|---|---|---|---|---|---|---|
| GPT-2 | 50.21 ±0.53 | **65.66** ±0.09 | **57.17** ±0.16 | **57.59** ±0.29 | 62.82 ±0.17 | 52.36 ±0.26 | 64.47 ±0.26 |
| BERT | **48.29** ±3.03 | 73.96 ±0.53 | **64.37** ±0.29 | **55.65** ±1.20 | **62.83** ±1.87 | 62.7 ±0.27 | 68.18 ±0.71 |
| RoBERTa | 54.78 ±0.51 | **72.83** ±0.46 | 64.63 ±0.1 | **60.67** ±0.25 | 67.87 ±0.35 | 64.98 ±0.34 | 71.73 ±0.17 |

Table 4: Average Spearman correlation performance of 5 repetitions of the local method ± standard deviation across these repetitions. The results in bold and black represent cases where the local method outperforms the global method with high probability and the results in red vice-versa.

| | Model | STS 2012 | STS 2013 | STS 2014 | STS 2015 | STS 2016 | SICK-R | STS-B |
|---|---|---|---|---|---|---|---|---|
| Baseline | GPT-2 | 9.3e-16 | 1.4e-120 | 1.5e-79 | 5.9e-92 | 1.5e-14 | 2.1e-121 | 3.7e-116 |
| | BERT | 2.5e-5 | 1.0e-4 | 1.1e-4 | 3.8e-5 | 5.3e-4 | 8.6e-5 | 1.1e-4 |
| | RoBERTa | 5.7e-6 | 3.5e-6 | 4.0e-6 | 5.9e-6 | 4.3e-6 | 5.8e-6 | 6.2e-6 |
| Global appraoch | GPT-2 | 0.56 | 0.59 | 0.51 | 0.57 | 0.58 | 0.56 | 0.56 |
| | BERT | 0.46 | 0.50 | 0.55 | 0.45 | 0.45 | 0.26 | 0.52 |
| | RoBERTa | 0.89 | 0.88 | 0.89 | 0.88 | 0.87 | 0.90 | 0.89 |
| Cluster-based approach | GPT-2 | **0.71** | **0.70** | **0.71** | **0.73** | **0.72** | **0.79** | **0.69** |
| | BERT | **0.75** | **0.71** | **0.73** | **0.76** | **0.72** | **0.79** | **0.76** |
| | RoBERTa | **0.91** | **0.90** | **0.91** | **0.92** | **0.91** | **0.95** | **0.92** |

Table 5: Isotropy of contextual word embeddings before and after enhancement with the global and local method.

### 4.1.2 GLUE & SuperGLUE classification tasks

In this section we address Claim 2 from Section 2. In Table 6 we report average scores (accuracy / Matthew's correlation) of the MLP classifier on the test set based on 5 repetitions. Each repetition, we re-ran the corresponding embedding enhancement method and randomly re-initialized and re-trained the MLP, accounting for both sources of variance.

It seems that the classifier trained on locally enhanced embeddings achieves the best scores on most of the tasks, however, due to the high uncertainty and small differences between methods, we cannot confidently argue that one method is better than the other. Due to this uncertainty, our results do not fully support the original findings as seen in Table 3 in the paper.

| | RTE | CoLA | SST-2 | MRPC | WiC | BoolQ | Average |
|---|---|---|---|---|---|---|---|
| Baseline | 54.7 ±1.4 | 7.4 ±16.5 | **84.0** ±0.3 | 66.8 ±0.7 | 53.3 ±5.7 | 62.3 ±0.05 | 54.75 ±4.1 |
| Global approach | 54.3 ±2.0 | 39.9 ±1.7 | 79.7 ±0.3 | 69.6 ±0.8 | 61.5 ±0.8 | **63.4** ±0.5 | 61.4 ±1.0 |
| Cluster-based approach | **55.1** ±1.6 | **40.1** ±1.8 | 83.7 ±0.8 | **70.2** ±1.2 | **61.9** ±0.8 | 62.7 ±0.6 | **62.3** ±1.1 |

Table 6: Results on classification tasks (BERT) in terms of accuracy (except for CoLA: Matthew's correlation). Results are based on averages and standard deviations of 5 runs on the official test set. In bold we mark the highest average score in each column.

### 4.1.3 Convergence time

In this section we address Claim 3 from Section 2. In Figure 2, we plot the per-epoch performance of the MLP for two SuperGLUE tasks on the validation split. Our results support the original claim, as the MLP converges to an optimum in only a few iterations when trained on enhanced embeddings, while the same does not hold for baseline embeddings.

### 4.1.4 Punctuation and stop word experiment

In this section we address Claim 4 from Section 2. In Figure 3, we plot the percentage of nearest neighbors from the same structural (syntactical) group, for baseline and enhanced embeddings. The results line up with the authors' results (Figure 3 in original paper) for BERT and RoBERTa embeddings, where the removal of dominant directions via the method decreases the percentage of neighbors from the same group. However, this does mostly not hold for GPT-2 embeddings in our reproduction.

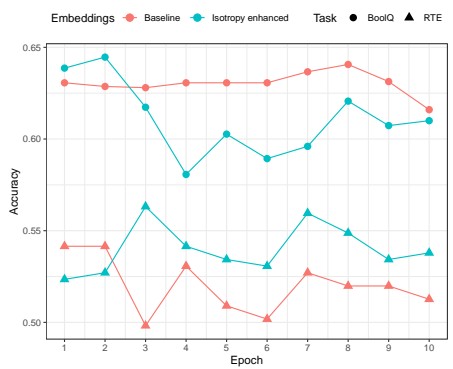
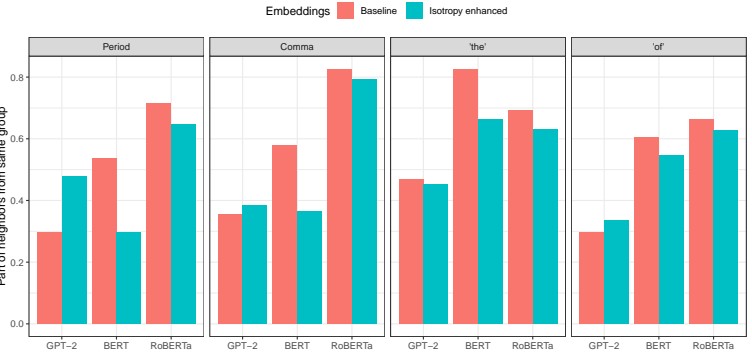

Figure 2: Impact of cluster-based enhancement on per-epoch performance.

Figure 3: Percentage of nearest neighbours that share similar structural and syntactic knowledge, before and after removing dominant directions.

### 4.1.5 Verb tense experiment

In this section we address Claim 5 from Section 2. In Table 7, we report the results of the corresponding experiment, described in Section 3.1. The results support the claim, as they are very similar to authors' results in Table 4 of the original paper.

| Model | Baseline | | | | Removed PCs | | | |
|---|---|---|---|---|---|---|---|---|
| | ST-SM | ST-DM | DT-SM | Isotropy | ST-SM | ST-DM | DT-SM | Isotropy |
| GPT-2 | 39.62 | 38.12 | 42.18 | 2.3e-05 | 5.06 | 5.56 | 5.43 | 0.708 |
| BERT | 13.43 | 13.69 | 14.04 | 2.41e-05 | 10.74 | 11.50 | 11.35 | 0.72 |
| RoBERTa | 6.20 | 6.39 | 7.09 | 6.2e-06 | 4.10 | 4.48 | 4.46 | 0.82 |

Table 7: Mean Euclidean distance of each occurrence of a verb to all other occurrences of the same verb with same tense and same meaning (ST-SM), the same tense and different meaning (ST-DM), and different tense but same meaning (DT-SM). It is desirable that DT-SM is lower than ST-DM.

### 4.1.6 Additional isotropy analysis

In this last section, we report the reproduction results of the additional isotropy analysis of the contextual models' embeddings. The results, analyzing the impact of number of clusters in K-Means and the layer-wise isotropy of the contextual models are seen in Tables 8a and 8b respectively. Our results support the original results, as seen in Tables 1 and 5 in the original paper.

## 5 Discussion

In general, many of the original authors' claims are supported by our experimentation. The achieved isotropy scores across the reproduced experiments are similar to the original ones, implying that the cluster-based method is working as intended. However, even in situations with seemingly no randomness (extracting baseline embeddings of datasets and evaluating isotropy), we could not perfectly reproduce the original results. This might imply discrepancies on hardware-level computation or due to different versioning of used libraries (e.g. Transformers). Consequently, this

Table 8: Additional isotropy analysis. In 8a, we report CWRs isotropy after clustering and zero-centering for different number of clusters ($k$). In 8b we report per-layer isotropy.

(b)

| Layer | GPT-2 | BERT | RoBERTa |
|---|---|---|---|
| 0 | 8.8e-03 | 4.7e-04 | 9.0e-03 |
| 1 | 9.4e-24 | 9.4e-06 | 2.5e-07 |
| 2 | **1.3e-24** | 1.0e-06 | 8.6e-10 |
| 3 | 5.9e-26 | 8.7e-05 | 4.2e-09 |
| 4 | 1.5e-27 | 7.4e-06 | 5.4e-12 |
| 5 | 2.9e-30 | 4.8e-06 | 4.9e-10 |
| 6 | 1.5e-32 | 3.8e-06 | 3.1e-10 |
| 7 | 1.3e-37 | 5.1e-06 | 1.3e-10 |
| 8 | 3.3e-45 | 1.1e-05 | 1.4e-10 |
| 9 | 5.0e-55 | 2.5e-05 | 1.4e-10 |
| 10 | 7.0e-34 | 4.3e-06 | 6.5e-11 |
| 11 | 1.9e-132 | 2.3e-07 | 1.4e-10 |
| 12 | 1.3e-126 | **4.9e-05** | **2.7e-06** |

(a)

| | GPT-2 | BERT | RoBERTa |
|---|---|---|---|
| Baseline | 1.27e-126 | 4.91e-05 | 2.69e-06 |
| k=1 | 3.62e-220 | 1.91e-05 | 0.015 |
| k=3 | 1.21e-73 | 1.15e-04 | 0.318 |
| k=6 | 3.36e-61 | 2.97e-03 | 0.512 |
| k=9 | **7.06e-54** | 0.148 | 0.549 |
| k=20 | 8.42e-101 | **0.265** | **0.579** |

perhaps implies that the local method is not robust enough to such variations, to consistently outperform the global method (e.g. in terms of Spearman coefficient performance on STS tasks), as originally claimed.

Similarly, for the classification tasks, after our own re-implementation, we found out that authors used Keras for the MLP classifier, while we used ScikitLearn (albeit with all hyperparameters set equivalently). This was another source of potential discrepancies, but the similar results reflect that this was not a real issue. A more likely reason for some differences in this experiment might be the fact that, while the authors stated that they re-trained the MLP multiple times before submitting and reporting the results of **the best classifier** (chosen by validation set performance), we opted for the more robust and less biased score estimation via averaging across multiple submissions and additionally estimating the errors of our estimates.

When it comes to Claims 3 and 5 from Section 2, our results fully support these claims, although again, we are unable to get exactly the same numbers, perhaps due to the reasons listed above or due to minor differences in implementation.

Finally, with the punctuation and stop word experiment, we were surprised by the fact that by removing local dominant directions of CWRs from the GPT-2 model, we actually *increased* the percentage of neighbors from the same structural group. Since the percentage of nearest neighbors with the same syntactical structure was relatively low to begin with in this case (compared to BERT and RoBERTa), we believe the dominant directions carried mostly semantic information, and by removing them, the syntactical information in the embeddings became more dominant.

## 5.1 Recommendations for further experimentation

Unfortunately, due to various limitations and our budget, we could not afford much additional experimentation beyond the scope of the paper. However, during our analysis, we came up with some ideas and experiments, which could be further looked into. We list some of these ideas the following.

Firstly, for the GLUE & SuperGLUE classification tasks, the authors first merge train and test splits and then run the embedding enhancement method and then train the MLP. In a practical scenario, where we would like to predict the class for a completely new data sample, repeating this whole process becomes computationally infeasible.

Therefore, the following experimental procedure, where the learning step is performed only once (and updated on a less regular basis), could be evaluated and compared to the original one:

1. Run the cluster-based method on contextual embeddings of the training set. Save the centroids of each cluster in original space as well as its corresponding top principal components to be removed.

2. Train the MLP on the enhanced embeddings.

3. At prediction time (for test data), extract the contextual embeddings of the new data sample. For each CWR, **enhance it by doing the following**: assign it to the nearest cluster, based on the saved centroids in step 1, then subtract the centroid and remove the corresponding PCs.

4. Pass the enhanced embeddings of the data sample to the MLP for prediction.

Other additional ideas include experimenting with different MLP architectures, or some of the remaining GLUE / SuperGLUE tasks, namely COPA, QNLI, QQP, etc. Additionally, using a different clustering algorithm or distance measure could prove to be beneficial.

### 5.2  What was easy

The explanations of the methods and experiments in the original paper were easy to follow. The cluster-based method relies on K-Means clustering and PCA, both of which we were already familiar with. The code present in the referenced repository was therefore easy to understand.

### 5.3  What was difficult

Some key implementation details of various experiments and hyperparameters of algorithms were not disclosed in the original paper, making exact re-implementation of the experiments more difficult. Even after receiving the necessary information, there were discrepancies in results which could not be attributed to randomness, differences in data, or some differences in implementation (assuming authors used the published code).

Due to some big datasets used in some experiments, we had to subsample the number of data samples to be able to run the described algorithms. Our system would in some cases completely freeze due our CPU usage reaching 100% because of PCA computations. Additionally, extracting embeddings, re-running the methods multiple times and performing expensive procedures such as bootstrap took a lot of time.

The most time-consuming step by far was estimating the performance and error of our estimates on GLUE and SuperGLUE classification tasks. In order to get test split results, one has to manually submit the predictions through the official website. This was an issue in our case due to the restrictions of submissions – a team is only allowed to make up to two submissions a day and six per month, which dragged out our collection of results.

### 5.4  Communication with original authors

We exchanged many e-mails with the main author of the paper, in order to enquire about various hyperparameters and other implementation details of each experiment and to ensure we set up our experiments the same way. The author was quite helpful and responsive. Unfortunately, we had to accept that some discrepancies between our results would still be present (see Sections 5 and 5.3 for our comments on these discrepancies), after much time spent attempting to reduce them.

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
