# OpenReview forum: "[Re] A Cluster-based Approach for Improving Isotropy in Contextual Embedding Space"
_ML_Reproducibility_Challenge/2021/Fall — RC2021_

### Official Review · Reviewer_vk4Z · 2022-03-04
**A Cluster-based Approach for Improving Isotropy in Contextual Embedding Space**

**Rating:** 9
**Confidence:** 4

**Review:**

NA

---

### Meta-Review · Program_Chairs · 2022-04-07

**Recommendation:** Accept
**Confidence:** 4

**Metareview:**

A clearly written and argumented paper that addresses and important and relevant problem in NLP. The way the claims are listed and verified is systematic and well thought out. The authors also fix the original code, which is useful given the edge case. Section 3.2 could be better structured, maybe using a table, since it's hard to keep track of which dataset is which. Section 4 is well-written and well argumented. The discrepancies in the claims that were identified were interesting as well.

---

### Decision · Program_Chairs · 2022-04-09

**Decision:**

Accept

**Comment:**

Following the recommendation of reviewers and meta-reviewer, the paper is accepted for ML Reproducibility Challenge 2021, and will be published in the upcoming special edition of ReScience Journal.